# The Diagnostic Deceiver: Radiological Pictorial Review of Tuberculosis

**DOI:** 10.3390/diagnostics12020306

**Published:** 2022-01-25

**Authors:** Sultan Abdulwadoud Alshoabi, Khaled M. Almas, Saif A. Aldofri, Abdullgabbar M. Hamid, Fahad H. Alhazmi, Walaa M. Alsharif, Osamah M. Abdulaal, Abdulaziz A. Qurashi, Khalid M. Aloufi, Kamal D. Alsultan, Awatif M. Omer, Tareef S. Daqqaq

**Affiliations:** 1Department of Diagnostic Radiology Technology, College of Applied Medical Sciences, Taibah University, Almadinah Almunawwarah 42353, Saudi Arabia; fhdhazmi@taibahu.edu.sa (F.H.A.); wsheref@taibahu.edu.sa (W.M.A.); oabdulaal@taibahu.edu.sa (O.M.A.); aaqurashi@taibahu.edu.sa (A.A.Q.); kmoufi@taibahu.edu.sa (K.M.A.); kamal_hihs@yahoo.com (K.D.A.); awatefomer222@hotmail.com (A.M.O.); 2Radiology Department, Al-Hawbany Hospital, Alhodeidah, Yemen; almass.khalid@yahoo.com; 3Radiology Department, Central Military Hospital, Sana’a, Yemen; aldofri@yahoo.com; 4Radiology Department, Rush University Medical Center, Chicago, IL 60612, USA; gabbaryy@gmail.com; 5Radiology Department, Faculty of Medicine, Taibah University, Almadinah Almunawwarah 42353, Saudi Arabia; dr.tareef@gmail.com

**Keywords:** miliary tuberculosis, tree-in-bud pattern, tuberculomas, tuberculous lymphadenitis, cavitary lesions, tuberculous spondylitis, tuberculous pericarditis

## Abstract

Tuberculosis (TB) is a bacterial infection with *Mycobacterium tuberculosis*; it is a public health problem worldwide and one of the leading causes of mortality. Since December 2019, the COVID-19 pandemic has created unprecedented health challenges and disrupted the TB health services, especially in high-burden countries with ever-increasing prevalence. Extrapulmonary and even pulmonary TB are an important cause of nonspecific clinical and radiological manifestations and can masquerade as any benign or malignant medical case, thus causing disastrous conditions and diagnostic dilemmas. Clinical manifestations and routine laboratory tests have limitations in directing physicians to diagnose TB. Medical-imaging examinations play an essential role in detecting tissue abnormalities and early suspecting diagnosis of TB in different organs. Radiologists and physicians should be familiar with and aware of the radiological manifestations of TB to contribute to the early suspicion and diagnosis of TB. The purpose of this article is to illustrate the common radiologic patterns of pulmonary and extrapulmonary TB. This article will be beneficial for radiologists, medical students, chest physicians, and infectious-disease doctors who are interested in the diagnosis of TB.

## 1. Introduction

Tuberculosis (TB) is a bacterial infection with *Mycobacterium tuberculosis*. TB is considered to be a public health problem and a leading cause of death worldwide [1,2]. In 2020, the COVID-19 pandemic created unprecedented health challenges and replaced TB as the first deadly infectious disease, thus disrupting the TB health services, especially in high-burden countries [2]. It is estimated that 30% of the world’s population is infected with TB [3]. Typically, infection occurs through prolonged or repeated exposure to *Mycobacterium tuberculosis* transmitted by airborne droplets from the coughs or sneezes of an actively infected person. The lung is the most common site of TB infection; however, it can involve any other organ in the body [1,3]. Approximately 90% of exposed individuals carry the bacteria but harbor the infection at a subclinical level with no clinical, radiological, or microbiological manifestations, a condition known as latent infection. Nearly 5% of infected patients develop active infection within the first two years; this category is called primary TB. The remaining 5% of infected individuals with effective immunity can control the primary infection, but viable *mycobacteria* remain dormant and reactivate; this is known as post-primary or reactivated TB [3,4].

The lungs are the primary affected organs; however, hematogenous dissemination from latent pulmonary focus can occur in other organs in the body, thus forming extrapulmonary TB. It causes destructive changes in the affected organs due to caseous granulomas, which are formed from the development of cell-mediated hypersensitivity. This process of infection can be confused with many benign or malignant neoplasms [5].

Clinical manifestations of TB include night fever and sweating, unexplained weight loss, loss of appetite, and fatigue. A persistent cough, with or without hematemesis; breathlessness; and chest pain when breathing are additional manifestations in highly contagious patients with pulmonary TB. However, clinical manifestations vary in extrapulmonary TB according to the infected organs; therefore, the diagnosis of extrapulmonary TB requires a high index of suspicion and further laboratory and radiological investigations [4,5]. The diagnosis of TB requires several tests to be performed. The smear test and culture of *Mycobacterium tuberculosis* are the gold-standard tests. However, they are expensive and deliver a late diagnosis. Latent TB can be diagnosed by using a tuberculin skin test or an interferon-gamma release assay, which have no role in the diagnosis of active TB. In active TB, nucleic acid amplification tests (NAATs) include DNA extraction and polymerase chain reaction (PCR) amplification; the new molecular diagnostic test Xpert mycobacterium tuberculosis/rifampicin resistance (Xpert MTB/RIF) assay, which can detect the *Mycobacterium tuberculosis* complex within two hours, and the histopathological examination of biopsy samples and medical imaging are required for evaluation. Recently, Lipoarabinomannan (LAM) has been suggested as a biomarker to diagnose TB based on its identification in urine [6,7]. In particular, extrapulmonary TB is an important cause of nonspecific clinical manifestations, such as a fever of unknown origin (FUO) in TB-burden regions. However, medical-imaging tests play an essential role in detecting tissue abnormalities and the early suspected diagnosis of TB in different parts of the body [8].

Due to the significant role of medical imaging in the diagnosis of TB, we introduce this radiological pictorial review to elucidate the common various radiological patterns of pulmonary and extrapulmonary TB. The available previous studies cover this topic focus only on one system or one part of the body. However, this study discusses the most common differencing patterns of TB with radiological images and analyzes the possible differential diagnoses in various body systems. This review will be significant for radiologists, chest physicians, and infectious disease doctors who are interested in diagnosing TB.

## 2. Discussion

Medical imaging plays a leading role in the detection of tissue changes and primary suspected diagnosis of TB. All of the conventional radiography, computed tomography (CT), ultrasonography, magnetic resonance imaging (MRI), and positron emission tomography CT (PET-CT) methods have a role in the diagnosis of TB and can assist the physicians in their diagnosis of a wide range of diseases in different parts of the body. Various types of TB, namely pulmonary, abdominal, scrotal, brain and spine, and musculoskeletal (MSK), can be diagnosed via a wide range of medical imaging modalities [9,10]. Here, we discuss the common radiological features of TB in the chest, the central nervous system (CNS), the abdomen, and the MSK using the most appropriate imaging modalities.

### 2.1. Pulmonary TB

Lungs are the most common organ affected and they are often the initial sites of TB infection [11]. Pulmonary TB can be divided into primary and postprimary (reactivation), each with appropriate radiological features that are overlap considerably. It can involve the lung and extrapulmonary organs in the chest, such as pleura or lymph nodes [12,13].

#### 2.1.1. Radiological Patterns of Primary TB

Primary TB is acquired by the inhalation of airborne *Mycobacterium tuberculosis* in non-previously exposed persons. It commonly affects children in endemic areas. However, it is now increasingly estimated in adults even in non-endemic areas. Primary TB can affect any part of the lung parenchyma, the lymph nodes (LNs), the tracheobronchial tree, and the pleura that form any of the following four patterns: gangliopulmonary TB, which is a characteristic of primary TB, miliary TB, TB pleuritis, and tracheobronchial TB [12,13]. Radiological patterns of primary TB include the following:Lymphadenopathy (Figure 1) is the most common radiological manifestation of primary TB. It is characterized by right paratracheal, hilar, or subcarinal enlarged LNs with or without parenchymal abnormalities. Lymphadenopathy is more likely to occur in children than in adults and may cause pulmonary atelectasis.

A CT is more effective than a chest X-ray (CXR) for the detection of LNs, which are characterized by a low density center and peripheral enhancement after contrast administration, which form the “rim sign” [1,13]. Bilateral hilar lymphadenopathy is an important differential diagnosis of many benign and malignant diseases, such as sarcoidosis and lymphoma [13]. 2.Parenchymal disease (Figure 2) is another radiological pattern of primary TB that most frequently occurs as consolidation distributed as segmental or lobar opacity; however, a lack of strong lobar predilection and cavitation is not a common feature of primary TB [1,13].

#### 2.1.2. Radiological Patterns of Postprimary TB

Post-primary TB occurs in previously sensitized patients either due to the reinfection or reactivation of a dormant bacilli from a primary infection, commonly occurring in adolescents and adults. It usually involves the posterior and apical segments of the upper lobes and the superior segments of the lower lobes. It is characterized by the liquefaction of caseous necrosis with the formation of cavities, progressive lung fibrosis and destruction, and bronchogenic spread [13,14]. Radiological images can suggest that TB is active by recognizing the following features: centrilobular nodules, bronchial wall-thickening, poorly defined nodule, thick-wall cavities, lobular consolidation, pleural effusion, miliary nodules, and necrotic enlarged lymph nodes [9,10,14]. However, thin-wall cavities, lung fibrosis, pleural thickening, calcified or non-calcified nodules, and lung parenchymal or pleural calcification may denote an old infection [9,15]. Consolidation (Figure 3) is considered to be one of the most common features of postprimary TB, which is usually focal, patchy heterogeneous, or poorly defined. It involves the apical and posterior segments of the upper lobes and the upper segments of the lower lobes [10]. Consolidation with ipsilateral enlarged hilar or paratracheal LNs could strongly suggest TB. CT is better able to detect small and subtle TB consolidations, which are usually peribronchial or subpleural and involve multiple lung segments [13].Cavitation (Figure 3 and Figure 4) is a common finding in postprimary TB, and it is characterized as being several centimeters in size with thick irregular walls. Cavities are often seen within consolidation and may persist after treatment predisposing to a bacterial or fungal superinfection or adjacent vascular erosion causing hemoptysis [1,10]. In postprimary TB, both consolidation and cavitation have a predilection for the apical and posterior segments of the upper lobes and the upper segments of the lower lobes [1,14]. This predilection of TB is attributed to the relative over-ventilation, high oxygen tension, and delayed lymphatic clearance in these regions [16]. Thick wall cavities are an important differential diagnosis of a pulmonary abscess, septic emboli, aspergilloma, granulomatosis with polyangitis (Wegener’s granulomatosis), lung malignancy, and others [17].Centrilobular nodules (Figure 4 and Figure 5) occur due to the communication of active TB with the bronchial tree resulting in endobronchial spread. It occurs in most cases of active TB. It appears as centrilobular nodules and a tree-in-bud sign on CT images [1,10]. The tree-in-bud pattern is seen on high-resolution CT images as 2–4 mm centrilobular nodules of soft tissue density that are connected to multiple branching linear structures of the similar caliber, arising from a single stalk. It commonly occurs in the endobronchial spread of TB and is highly suggestive of active TB. However, a tree-in-bud is a CT manifestation of the diverse entities of lung diseases, including TB, cytomegalovirus, respiratory syncytial virus, obliterative bronchiolitis, diffuse panbronchiolitis, cystic fibrosis, airway-invasive aspergillosis, allergic bronchopulmonary aspergillosis, and pulmonary metastasis [18].

#### 2.1.3. Radiological Patterns of Both Primary and Postprimary TB

Some radiological patterns occur in both primary and postprimary TB with variation in the percentages of occurrence.
Miliary TB (Figure 6 and Figure 7) appears as innumerable small (1–3 mm) granulomas with random distribution in the lungs and other organs with a predominance to the lung bases due to the gravity-dependent high blood flow. It occurs due to the hematogenous dissemination of mycobacterium tuberculosis bacilli, especially in immunocompromised patients and children [1,13]. Miliary TB is a significant differential diagnosis of pulmonary metastasis from thyroid cancer or others, even in children [19].Pleural effusion (Figure 8) can occur in both primary and postprimary TB; however, it is more common in primary TB, in which it manifests as a unilateral free pleural effusion with no loculations. It is more common in adolescents and adults than in children. It occurs 3–6 months after infection due to a hypersensitivity reaction to mycobacterial antigens and tests negative for bacilli. However, it is usually small and loculated with associated parenchymal lesions in postprimary TB. It originates from the rupture of a cavity into the pleural space, so the cell culture is usually positive for bacilli [13,20].Tuberculoma is a well-defined smooth-margin granulomatous nodule or mass up to 5 cm in size, which can be either solitary or multiple and may occur in both primary and postprimary TB. It is more commonly a result from healed primary TB, and the majority remains stable in size and may develop nodular or diffuse calcification or cavitation [13]. Pulmonary tuberculoma can mimic and should be a differential diagnosis of lung cancer [21].Tracheobronchial TB can affect the tracheobronchial tree, as well as to the lung, which may cause tracheobronchial stenosis and severe complications. CTs, bronchoscopies, and microbiological investigations are diagnostic tools used to confirm diagnoses and evaluate tracheobronchial stenosis [22].

Even after full treatment of pulmonary TB, complications and residual changes may affect quality of life; these complications can be misinterpreted as other active diseases, causing diagnostic pitfalls. TB complications in the lung parenchyma include tuberculoma, thin-walled cavities, cicatrisation collapse (Figure 9), and lung cancer, although this is rare. Complications in the airway include bronchiectasis (Figure 10) and tracheobronchial stenosis. Extrapulmonary complications include pleural thickening and calcification, fibrothorax, bronchopleural fistula, and pneumothorax (Figure 11). Vascular complications include Rasmussen aneurysm, calcified mediastinal lymph nodes, fibrosing mediastinitis, and constrictive pericarditis [23].

### 2.2. Extrapulmonary TB

#### Central Nervous System (CNS)

TB can involve the CNS in 10% of cases, and it is the most devastating form of systemic TB, due to its serious neurological complications and sequelae, with high mortality rate [24,25]. In the majority of cases, infection comes via hematogenous spread from a pulmonary focus and is rarely via a direct spread from paranasal sinuses, mastoids, or orbit. It can present in various clinical conditions with clinical manifestations that are similar to other diseases, including bacterial meningitis, intracranial hemorrhage, and primary or secondary neoplasms. In addition to the rarity and unfamiliarity, the diagnosis of extrapulmonary TB is difficult. However, radiological patterns can be identified well by using an MRI, which plays a crucial role in the suggested diagnosis of TB [24,26]. It can affect leptomeninges (pia matter and arachnoid), causing tuberculous leptomeningits or pachymeninges (dura matter), thus resulting in pachymeningitis [25,26]. TB can affect the brain parenchyma, causing tuberculoma, miliary tuberculomas, cerebritis or abscesses. Moreover, it can affect the spinal cord, causing spinal tuberculoma, tuberculous myelitis, arachnoiditis, or spondylodiscitis (Potts disease) [27]. Tuberculous leptomeningitis (Figure 12) is considered to be one of the most common types of neurotuberculosis. It is frequently seen in children and is associated with complications. On an MRI, it appears as diffusely enhancing exudates and leptomeningeal enhancement with predilection where basal cisterns are involved. Magnetization transfer MRI (MT-MRI) is highly valuable for the diagnosis of mild cases of meningitis [25,27]. Tuberculous leptomeningitis has a wide differential diagnosis, including pyogenic leptomeningitis, fungal leptomeningitis, and leptomeningeal carcinomatosis [28]. It can also mimic sarcoidosis [29].Tuberculoma (Tuberculous granuloma; Figure 12 and Figure 13) is one of the most common brain parenchymal tuberculous lesions and can be solitary or multiple anywhere within the brain. It is commonly seen at the corticomedullary junction and periventricular region as a result of hematogenous dissemination. MRI features of tuberculoma vary according the stage of maturation (stage 1, non-caseating; stage-2, caseating granuloma; stage 3, caseating granuloma with central liquefaction; and stage 4, calcified granuloma). Radiologically, each stage of brain tuberculoma can mimic a wide variety of differential diagnoses, such as neurocysticercosis, fungal granulomas, pyogenic abscess, metastasis, glioma, lymphoma, and toxoplasmosis [25,27]. Ring-enhancing lesions of the brain may form a diagnostic dilemma [30].Miliary tuberculomas (Figure 13) occur due to the haematogenous dissemination of *Mycobacterium tuberculosis* bacilli, which are usually lung focused and occur especially in immunocompromised patients. They appear as innumerable small (2–3 mm) non-caseating granulomas with random distribution in the brain with a predominance to a gray-white matter junction as a result of hematogenous dissemination [25,26]. Miliary TB usually strongly mimics brain metastasis [31].Tuberculous cerebritis (Figure 14) is an infection of a focal area of a brain parenchyma that appears on MRI as an area of swelling with an alteration to the signal intensity of the gyri. The involved gyri appear with a low signal intensity on T1WIs and high signal on T2WIs with patchy enhancement after contrast administration [25,26]. Neuroimaging using a CT and MRI plays a crucial role in early diagnosis and avoids the worst complications of cerebritis. Tuberculous cerebritis is characterized by a small tuberculous granuloma and appears as intense areas of patchy enhancement with edema which mimics infarction those can be differentiated by diffusion-weighted MRI [32].Tuberculous abscesses (Figure 15, Figure 16 and Figure 17) are an infrequent pattern of brain TB, which are present in immunocompromised and elderly patients. They occur either due to the progression of cerebritis or the liquefaction of tuberculoma. On a brain CT, it occurs as a low density area with ring enhancement after contrast administration and the surrounding low density area of edema. On a brain MRI, it appears as a circular or elliptical area of low intensity with ring enhancement after contrast administration and surrounding low intensity area of edema on T1WIs and high intensity area with a high intensity surrounding edema. Tuberculous abscess is a difficult differential diagnosis of a pyogenic abscess, pilocystic astrocytoma, and other cystic lesions. MR spectroscopy may help in differentiating tuberculous from a pyogenic abscess [27,33].Tuberculous spondylitis (Pott’s disease; Figure 18 and Figure 19) is a tuberculous infection of the spine that can affect any age and most commonly affects the lower thoracic and lumbar, followed by the cervical spine. It usually affects multiple contiguous vertebrae with paraspinal extension, causing subdural or epidural abscess formation and resultant spinal cord compression, which is a leading cause of paraplegia [27]. CT demonstrates the extent of bone involvement, while MRI demonstrates spinal-cord and soft-tissue involvement. Tuberculous spondylitis appears as a well-defined paraspinal abnormal signal intensity, as a thin wall abscess, or a combination of intraosseous and soft tissue abscesses. The spread of infection is sub-ligamentous beneath the anterior longitudinal ligament involving multiple vertebrae with high signal intensity on T2WIs. Pott’s disease is a difficult differential diagnosis with a spinal pyogenic infection. MRI demonstrate have 100% sensitivity, 80% specificity, and 90% accuracy in differentiating tuberculous from pyogenic spinal infection [34].

Other rare forms of CNS TB include pachymeningitis, tuberculous encephalopathy in children, cerebral vasculitis, tuberculoma in plaque, tuberculous transverse myelitis, spinal tuberculous arachnoiditis, tuberculous hypophysitis, tuberculous otitis media, orbital TB and others [27,35].

### 2.3. Cardiovascular System (CVS) TB

The CVS is one of the most common sites (after the CNS) of the extrapulmonary TB, which is an affected system with an unfavorable prognosis. It affects the heart through lymphatic spread, which comes directly from the lungs or the pleura or hematogenous spread in miliary TB. TB can affect the pericardium, the myocardium, and the aorta. Pericardial involvement is relatively common, especially in patients with acquired immunodeficiency syndrome (AIDS). However, tuberculous myocarditis and aortitis are relatively rare manifestations. Transthoracic echocardiography (TTE) and cardiac MRI are imaging methods for the evaluation of the heart and pericarditis. CT allows the evaluation of the aorta for aortitis and the localization of aneurysms [36].

Pericardial involvement in TB can occur as pericardial effusion, constrictive pericarditis, or both. In developing countries, more than half of large pericardial effusions are tuberculous in origin, and TB is the most common cause of constrictive pericarditis. TTE is the initial imaging method to detect suspected pericardial effusion. CTs and MRIs are useful in detecting lobulated effusions and pericardial thickening. CTs have the advantage of detecting pericardial calcifications, which are a frequently found in constrictive pericarditis [37,38].
Pericardial effusion (Figure 8b) occurs in 85% of tuberculous patients with AIDS [36]. This may be associated with pericardial thickening and determining the cause of effusion by imaging is difficult [38]. Pericardial effusion has a wide range of differential diagnosis, including malignancy, infection, uremic, iatrogenic, idiopathic, heart failure, and autoimmune [39].Constrictive pericarditis may occur after the persistence of pericardial inflammation for years, leading up to a cardiac restriction. It often mimics other pathologies, such as restrictive cardiomyopathy, and other causes of heart failure, including pulmonary hypertension and cardiomyopathy. A cardiac MRI is the imaging modality of choice to confirm the diagnosis and to differentiate it from other pathologies [40]. TB is the leading cause of constrictive pericarditis in developing countries; however, it is a minor cause in developed countries [41].

TB rarely involves the myocardium and endocardium, and a left ventricular tuberculoma was reported as a rare case in a patient with human immunodeficiency virus (HIV) [42].

TB can simultaneously affect multisystems, such as the heart, blood vessels and brain causing disastrous conditions [43].

### 2.4. Abdominal TB

TB can affect any organ or tissue in the abdominal cavity and can be mistakenly diagnosed as other inflammatory or neoplastic conditions. The most common affected organs are lymph nodes, the genitourinary system (GUS), the peritoneal cavity, and the gastrointestinal tract (GIT). The spleen, hepatobiliary, pancreas, and adrenals are rarely affected. However, these organs are more likely affected in HIV patients and in miliary TB [44]. Tuberculous infection can be acquired through the dissemination of primary pulmonary TB in children, swallowing of infected sputum in an active pulmonary TB, hematogenous spread via active pulmonary focus, or miliary TB. Furthermore, infection can occur from dissemination from infected adjacent organs, such as the fallopian tubes, or through bile from liver tuberculoma, or via lymphatic spread from the mesenteric lymph nodes to the intestine [45].
Tuberculous lymphadenitis (Figure 20) is one of the most common radiological manifestation of abdominal TB which frequently involves multiple groups such as mesenteric and upper paraortic LNs. The majority of patients have enlarged lymph nodes with a low-attenuation center and peripheral-enhancing rim, which is characteristic of TB [45,46]. LNs may show peripheral rim enhancement, inhomogeneous, homogeneous, or no enhancement after contrast administration. Other patterns of LNs include conglomerate LNs with areas of necrosis, more than three enlarged or normal homogenous LNs in one section, or calcified LNs [45]. TB can mimic several other conditions, such as lymphoma, amebiasis, Crohn’s disease, and adenocarcinoma [47].Tuberculous peritonitis is the most common clinical presentation of abdominal TB. It involves the peritoneal cavity, mesentrium, and omentum. Peritonitis is believed to be originated from hematogenous spread or secondary to LN rupture, gastrointestinal dissemination, or fallopian-tube involvement [46,47]. Tuberculous peritonitis has three radiological patterns:Wet peritonitis (Figure 21) is the most common type that is characterized by free or loculated ascites with or without peritoneal thickening [46,47].Fibrotic peritonitis (Figure 22) is characterized by remarkable omental and mesenteric thickening forming cake-like masses with bowel loops enlargement and matting that can be seen by CT or ultrasound. [46,47].Dry peritonitis is characterized by mesenteric thickening with caseous nodules and fibrous adhesions [46,47].

Radiological features of tuberculous peritonitis can mimic nontuberculous infectious peritonitis, peritoneal carcinomatosis, peritoneal pseudomyxoma, and mesothelioma [46,47].

### 2.5. Gastrointestinal Tract (GIT) TB

GIT TB can occur with a pulmonary infection or as a primary infection without pulmonary involvement from the ingestion of infected milk products. TB can affect any part of the GIT from the mouth down to the anal canal; however, the ileocecal region is the most common affected site [48,49]. GIT TB can occur as (1) an ulcerative type, forming single or multiple mucosal ulcers in the GIT commonly in the jejunum and ileum; (2) an ulcero-hypertrophic type, forming a thickening and ulceration of the intestinal wall; and (3) a hypertrophic type, forming scarring and fibrosis commonly in the ileum and cecum [50].

Medical-imaging methods using a CT, capsule endoscopy, balloon enteroscopy, adenosine deaminase (ADA) in ascites, TB-PCR, GeneXpert, and laparoscopy are used to diagnose abdominal TB [49]. It is a significant medical diagnostic challenge worldwide and the diagnosis is usually difficult because it can mimic malignancies and inflammatory bowel diseases [48,51].

### 2.6. Genitourinary TB

The genitourinary system is one of the most common systems involved with TB infections and forms nearly 20% of extra pulmonary TB. It usually results from hematogenous seeding of *Mycobacterium tuberculosis* [52]. TB can affect both the renal parenchyma or any part of the collecting system from the pelvicalyceal system to the urethra occurring in nonspecific clinical features such as frequency, dysuria, hematuria, and flank pain. TB should be considered in patients with sterile hematuria and persistent cystitis. A CT provides high- quality images to diagnose urinary tract infections including TB, and MRI is used as a problem solving imaging modality if a CT is not diagnostic [53].

Radiological features of renal TB are varied and depend on the stage of a TB infection, which can present as pyelonephritis with total or partial swelling of the kidney, cortical scarring, cortical granulomas (Figure 23) and parenchymal calcifications (Figure 23d). TB infections of the collecting system vary from papillary necrosis at an early stage, and multifocal strictures, hydronephrosis, and dystrophic calcification to putty kidney (auto nephrectomy) at the kidney’s end-stage [54,55]. At each stage, renal TB can mimic other renal infections and even tumors. However, it can also be a differential diagnosis of renal cell carcinoma [56].

### 2.7. Musculoskeletal (MSK) TB

TB can affect any part of the musculoskeletal (MSK) system, and symptoms of TB may be insidious, causing TB not to be considered. CT and MRI imaging modalities can bring suspicion to TB diagnoses [57]. MSK TB has a wide range of radiological features that can mimic many pathologies all over the body. However, radiological assessment is often the first step in the diagnosis of MSK TB [58]. The most common type of MSK TB is tuberculous spondylodiscitis (Potts disease), which accounts for 50% of MSK TB [57,58] (Figure 18 and Figure 19). Osteoarthritis TB affects joints or bones. It usually affects long weight-bearing bones and sometimes affects the ribs. Articular TB usually presents as monoarthritis in the knee (Figure 24) or the hip joints. However, sacroiliac, and sternoclavicular joints are also sometimes affected. The predilection of TB to affect the vertebrae and the large joints is due to the rich blood supply to the vertebrae and the growth plates of the long bones [59]. TB arthritis presents as a slow progressive destructive monoarthritis, so the diagnosis is delayed due to the indolent onset and low clinical suspicion [60].

On medical imaging, TB may initially show soft tissue swelling and later progress to periosteal thickening, osteopenia, periarticular bone destruction, and cold abscesses, and fistulae may develop in late cases. MSK TB is a differential diagnosis of a wide bone lesion. At the early stages, it is often misdiagnosed as traumatic lesions, degenerative joint disease, gout, pseudo gout, rheumatoid arthritis, or pigmented villonodular synovitis. High suspicion of TB is required, and a final diagnosis can be carried out by using arthrocentesis and a mycobacterial culture; in addition, a synovial biopsy is often needed [59,60].

## 3. Limitations

This review is limited in that it does not cover all of the radiological features of TB, especially in the renal system due to the paucity of the available images in our centers. In the case of a brain tuberculoma, there were no available images for the four stages. There are also no available images for dry peritonitis.

## 4. Conclusions

TB is an important cause of nonspecific clinical and radiological manifestations, and it can masquerade as any benign or malignant medical case. Pulmonary TB can mimic any acute or chronic lung infection and even pulmonary carcinoma and metastasis. TB can masquerade as benign or malignant acute or chronic brain and spinal lesions. TB can cause ambiguous abdominal clinical and radiological manifestations, causing disastrous conditions and diagnostic challenges. Clinical manifestations and routine laboratory tests have limitations in directing physicians to diagnosing TB. It can appear similar to any acute or chronic disease of the kidney and genitourinary system. TB is an important differential diagnosis to most of acute and chronic lesions in the joints and bones. Medical-imaging tests play an essential role in detecting tissue abnormalities and the early suspected diagnosis of TB in different body parts. Radiologists and physicians should be familiar with and aware of the radiological manifestations of TB to contribute to the early suspicion and diagnosis of TB. This review highlights the frequent radiological manifestations of TB that should be known by any physician.

## Figures and Tables

**Figure 1 diagnostics-12-00306-f001:**
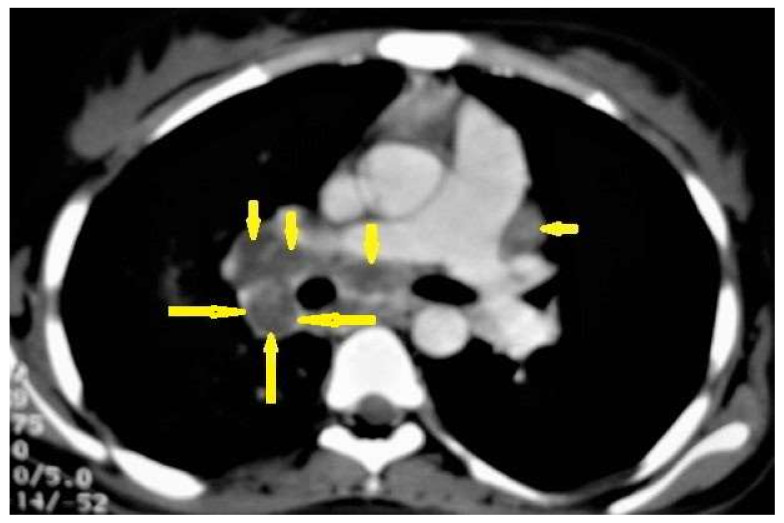
Primary TB in an 18-year-old man. Axial mediastinal-window CT image shows multiple enlarged mediastinal lymph nodes (short arrows), and right hilar lymph nodes are characterized by central low density and peripheral enhancement after contrast administration forming the rim sign (long arrows).

**Figure 2 diagnostics-12-00306-f002:**
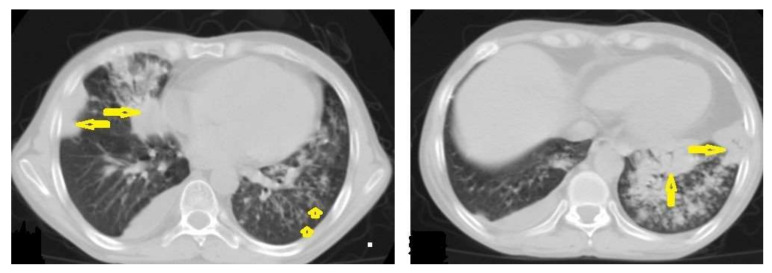
Active TB in a 27-year-old man with extensive endobronchial spread. Selected axial images of chest CT show extensive endobronchial spread characterized by patchy consolidations (long arrows) in the right middle and left lower lobes with tree-in-bud nodules (short arrow) more involved the left lung. Mild right pleural effusion and mild pericardial effusion appear in both images.

**Figure 3 diagnostics-12-00306-f003:**
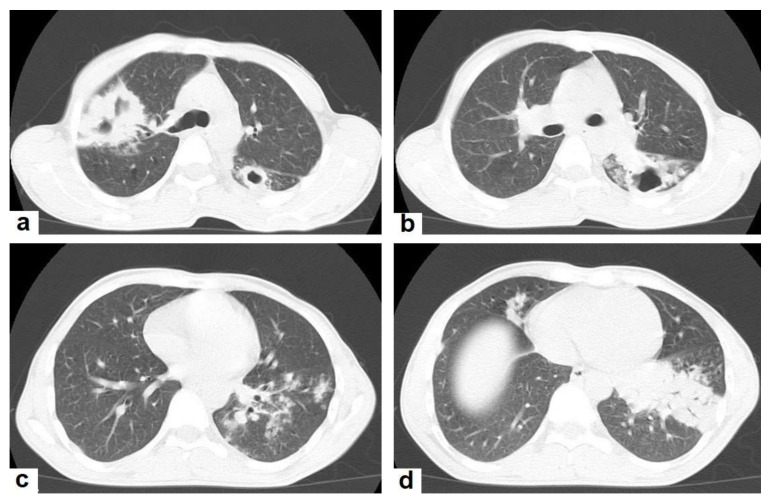
Post primary TB in a 45-year-old male with cough and hemoptysis. Axial images of chest CT show (**a**) cavitary lesions in the right upper lobe and upper segment of the left lower lobe surrounded with consolidation, (**b**) cavitary lesion in the upper segment of the left lower lobe with thick irregular wall surrounded by patchy ground glass opacities, (**c**) centrilobular nodules and tree-in-bud nodules, and (**d**) consolidation in the left lower lobe with air-bronchograms.

**Figure 4 diagnostics-12-00306-f004:**
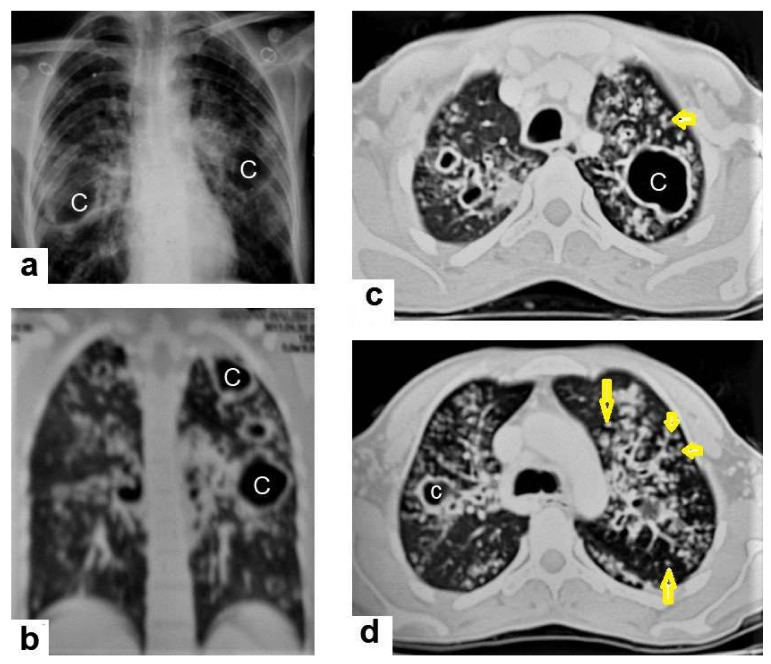
Reactivated TB in a 34-year-old women. (**a**) Chest radiograph shows extensive reticular shadowing and multiple cavitary lesions in both lungs. (**b**) Coronal reconstruction CT shows multiple cavitary lesions (**c**) in both lungs, the large two in the posterior segment of the upper lobe, and in the apical segment of the lower lobe of the left lung. (**c**,**d**) Axial CT images of the lung show multiple cavitary lesion, the largest in the posterior segment of the left upper lobe, with multiple centrilobular (long arrows) and tree-in-bud (short arrows) appearance.

**Figure 5 diagnostics-12-00306-f005:**
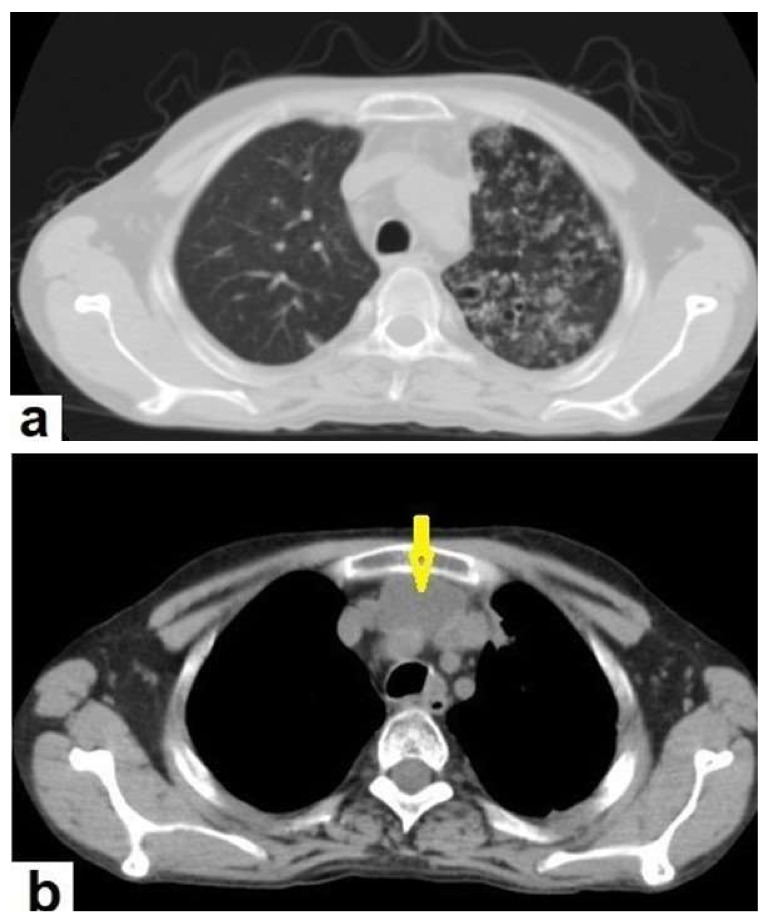
Active TB in a 27-year-old man (same case of Figure 2) with endobronchial spread. (**a**) Axial image of chest CT shows endobronchial spread characterized centrilobular and tree-in-bud nodules involved the upper lobe of the left lung. (**b**) CT axial image with mediastinal window shows large necrotic mediastinal lymph node measures 21 mm × 20 mm (arrow) in the anterior mediastinum.

**Figure 6 diagnostics-12-00306-f006:**
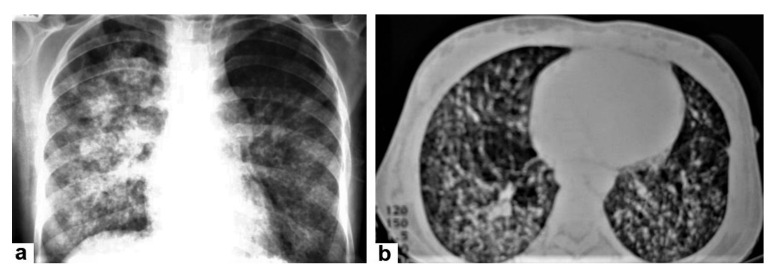
Miliary TB in an 18-year-old man (same patient of Figure 1). (**a**) Chest radiograph shows superimposed innumerable small nodules in lungs right greater than the left predominantly affecting lung bases due to gravity-dependent high blood flow. (**b**) Axial chest CT shows innumerable small (1–3 mm) nodules with random distribution in both lung fields.

**Figure 7 diagnostics-12-00306-f007:**
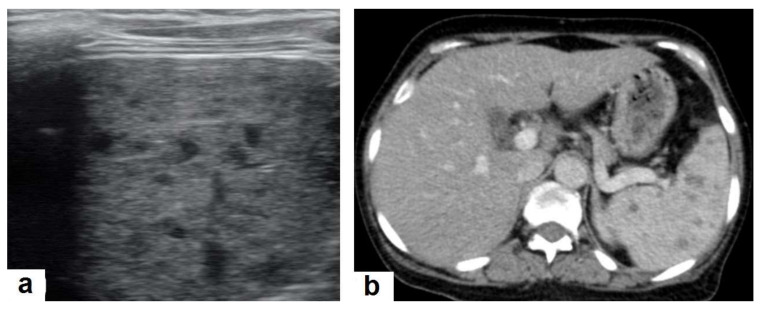
Miliary TB in an adult female with fatigue and loss of appetite. (**a**) Splenic ultrasonography shows multiple small hypoechoic granulomas with random distribution in the spleen. (**b**) Axial abdominal CT shows small non-enhancing granulomas with random distribution in the spleen clearly seen during the portal venous phase with multiple enlarged lymph nodes around the portal vein.

**Figure 8 diagnostics-12-00306-f008:**
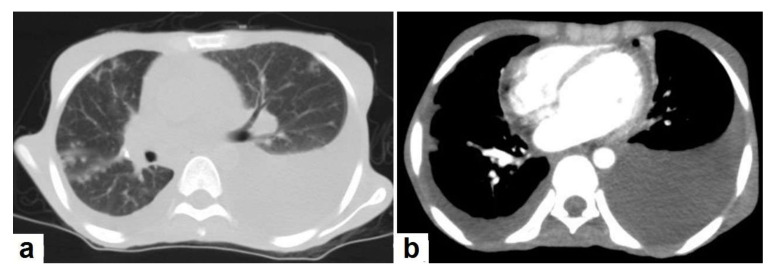
Active TB in a 27-year-old man (same case of Figure 2 and Figure 5). (**a**) Axial image of chest CT shows moderate left and mild right pleural effusion with discrete pulmonary nodules in active TB. (**b**) Axial image of the mediastinal window shows moderate left and mild right pleural effusion with mild pericardial effusion.

**Figure 9 diagnostics-12-00306-f009:**
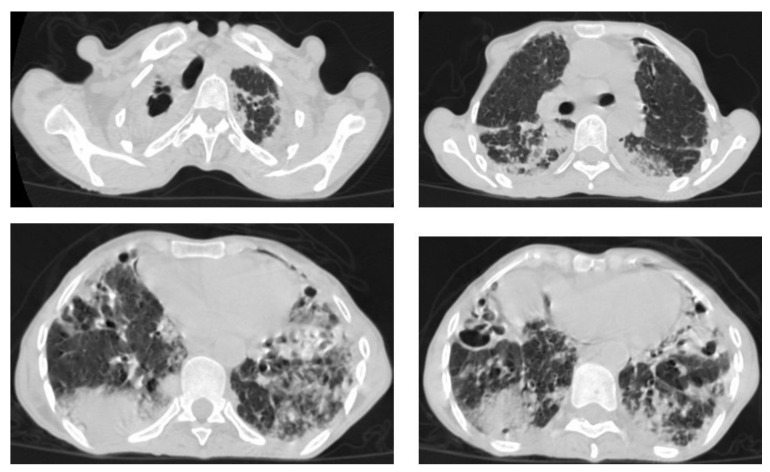
Old reactive TB with superimposed infection in a 75-year-old female with chronic cough and no history of previous medication. Selected axial images of lung CT show prominent fibrotic changes with scarring traction bronchiectasis and decreased volume in the apical and posterior segment of the upper lobes and architectural distortion of the lung parenchyma caused by cystic bronchiectasis predominantly involve bilateral lower lobes, ligula, and right middle lobe. Patchy consolidative areas and ground glass opacities are signs of active infection.

**Figure 10 diagnostics-12-00306-f010:**
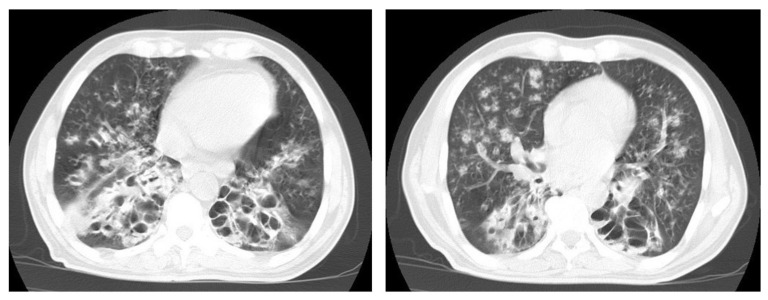
Reactivation of TB in an adult man with chronic cough. Selected axial images of lung CT show cystic bronchiectasis, due to old infection, with peribronchial thickening and air fluid level involving both lower lobes. Scattered tree in bud pattern and infected cystic bronchiectasis indicate active TB.

**Figure 11 diagnostics-12-00306-f011:**
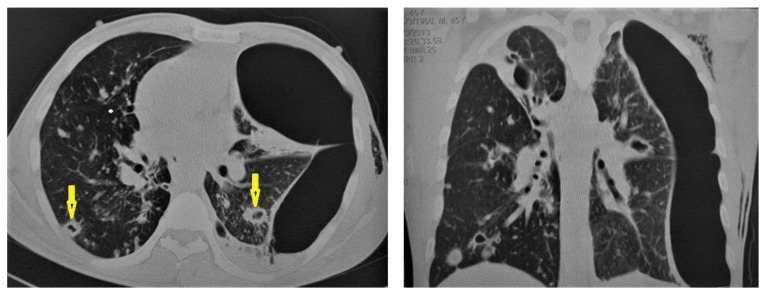
Post-primary-TB in a 45-year-old man. Selected axial and coronal reconstruction images of chest CT show multiple thick-walled cavitary lesions (arrows) in both lungs, with left pneumothorax as a complication of TB. Surgical emphysema in the left chest wall, due to chest-tube insertion.

**Figure 12 diagnostics-12-00306-f012:**
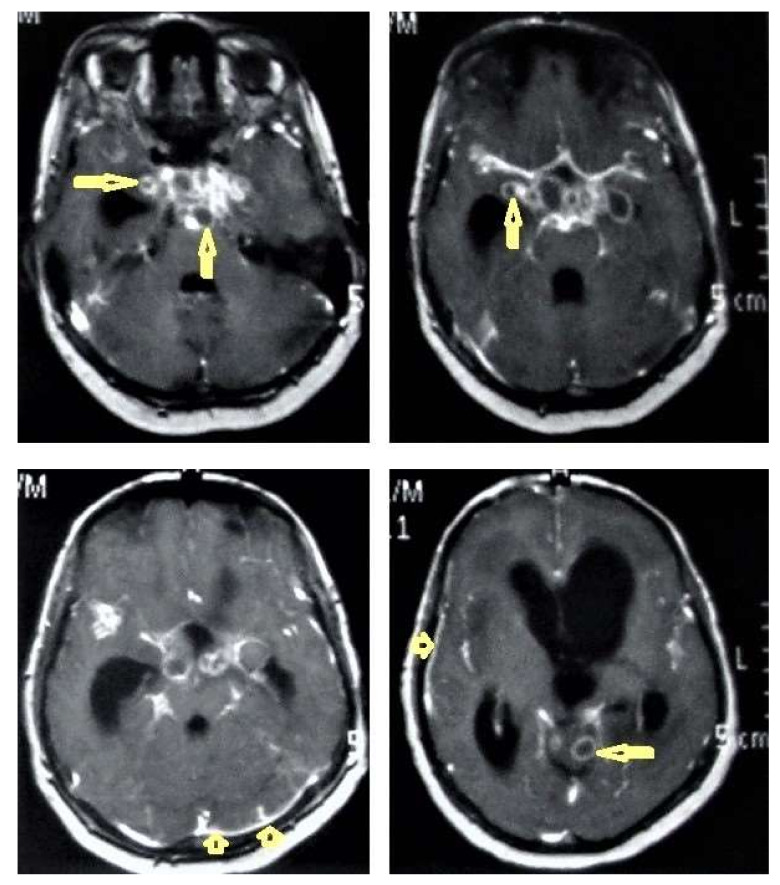
Tuberculous leptomeningitis. Selected axial T1-weighted images of brain MRI post-contrast administration show diffuse leptomeningeal enhancement (short arrows) with predilection to involve the basal cisterns complicated with hydrocephalus. Multiple ring-enhancing lesions (long arrows) in different regions of the brain.

**Figure 13 diagnostics-12-00306-f013:**
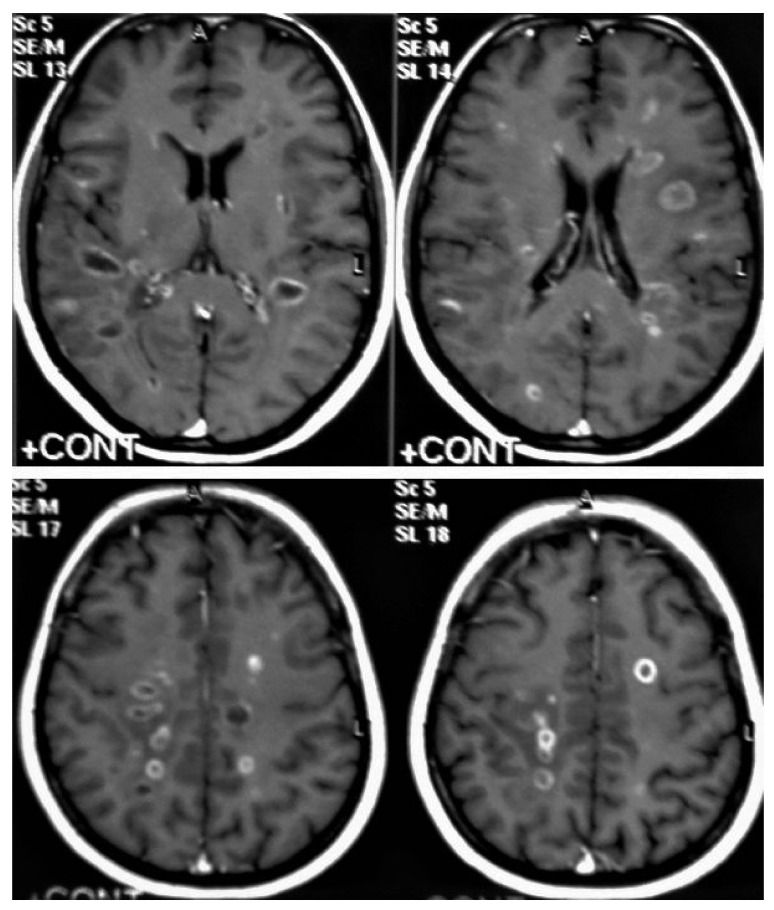
Multiple tuberculomas in an adult man. Selected MRI axial images post-contrast T1-weighted images show multiple small ring-enhancing lesions distributed in both cerebral hemispheres, with no significant surrounding cerebral edema, as is consistent with cerebral tuberculomas.

**Figure 14 diagnostics-12-00306-f014:**
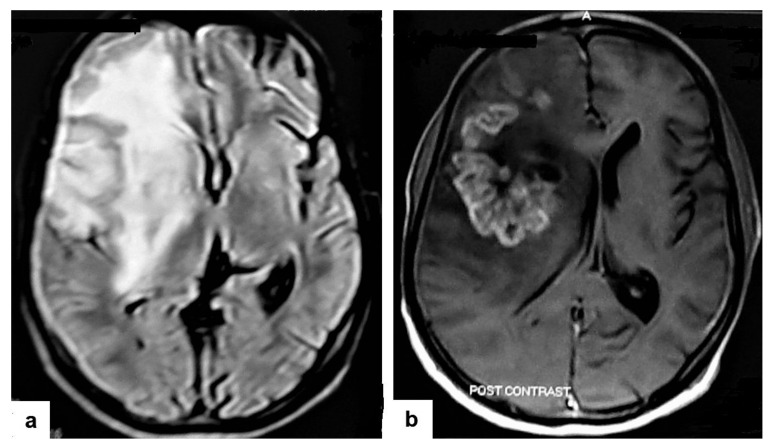
Tuberculous cerebritis in an adult man with fever and seizures. (**a**) Axial FLAIR-weighted images of brain MRI show high signal intensity in the right parietal and frontal lobes causing mass effect compressing the right lateral ventricle with mild shifting of the midline to the contralateral side. (**b**) Axial brain T1-weighted images post-contrast administration show intense serpentine (gyriform) enhancement in the right parietal and frontal lobes, with significant surrounding edema, suggesting tuberculous cerebritis.

**Figure 15 diagnostics-12-00306-f015:**
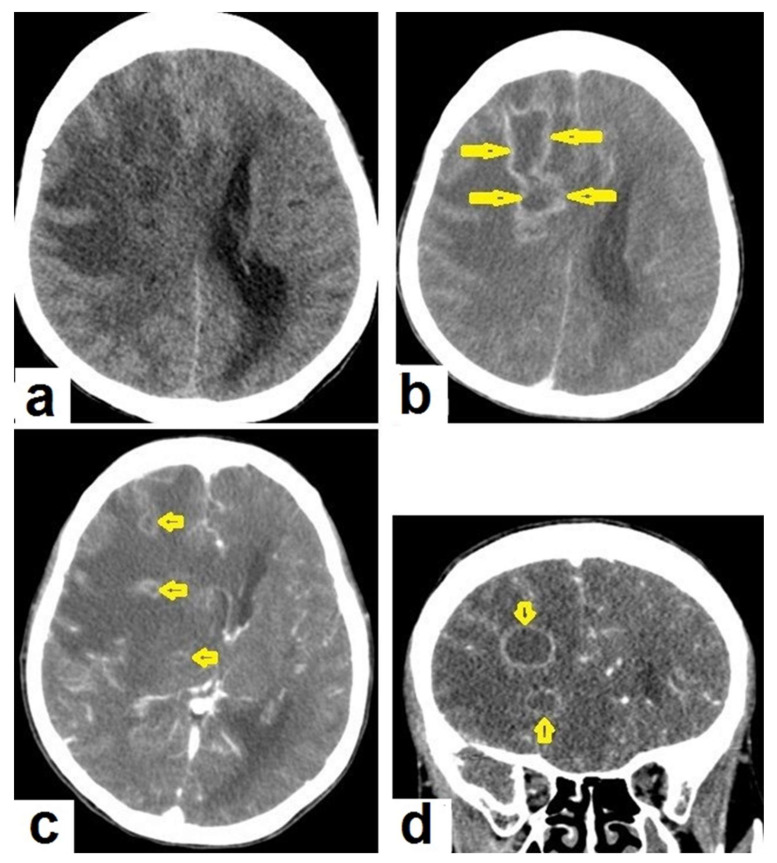
Tuberculous abscesses and tuberculomas in a 70-year-old woman with disturbance of consciousness and long history of headache. Selected axial computed tomography (CT) images of the brain (**a**) axial non-enhanced CT show significant vasogenic edema in the frontal and parietal lobes of the right cerebral hemisphere, with severe mass effect manifested as loss of the cortical sulci, total effacement of the right lateral ventricle and mild midline shift to the left side. (**b**) Contrast-enhanced CT (CECT) shows loculated rim-enhancing lesion centered in the right frontal lobe measures about 39 mm × 16 mm × 12 mm, suggesting cerebral abscess (arrows). (**c**) Axial CECT shows multiple ring-enhancing lesions in the right cerebral hemispheres and right basal ganglia (short arrows) suggesting of cerebral abscesses or tuberculomas. (**d**) Coronal CECT shows multiple ring-enhancing lesions in the right cerebral hemispheres (short arrows), suggesting cerebral tuberculomas or abscesses.

**Figure 16 diagnostics-12-00306-f016:**
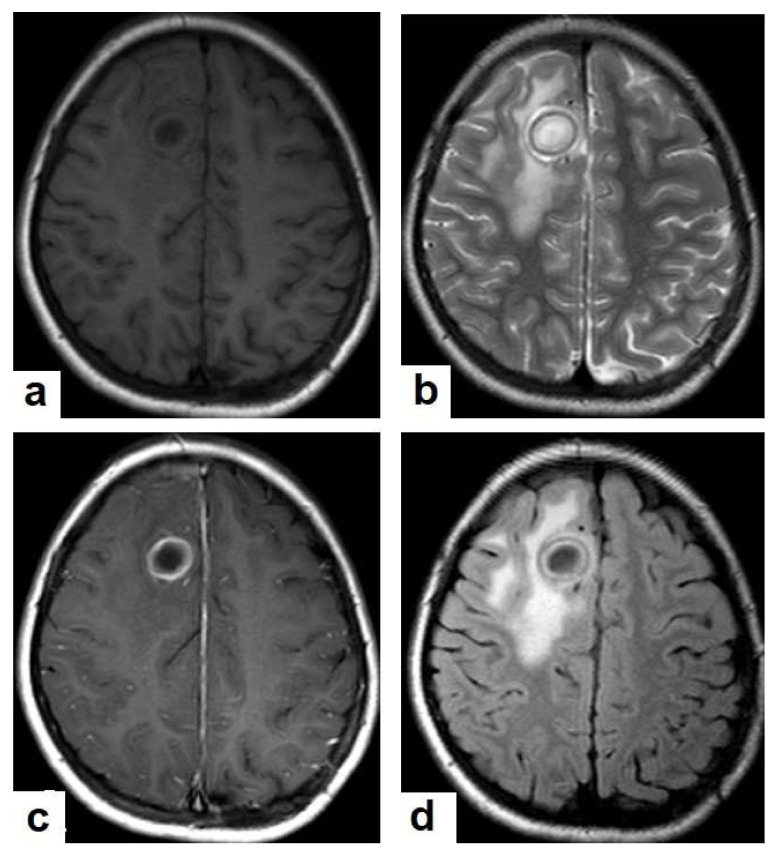
Tuberculous abscess in the right cerebrum of a 13-year-old female. Selected images of brain MRI (**a**) T1-weighted image (WI), (**b**) T2-WIs, (**c**) T1-WI with contrast, and (**d**) FLAIR-WI. The images show a well-defined lesion in the right cerebral hemisphere with low-signal-intensity content and a high-signal-intensity capsule on T1WI, which appear as high-signal-intensity content and a low-signal-intensity capsule on T2WI (**b**); there is marginal enhancement on T1WI with contrast administration (**c**), and there are low-signal-intensity contents and a high-signal-intensity capsule on FLAIR (**d**). Obvious grade-2 vasogenic oedema around the lesion on T2WI and FLAIR.

**Figure 17 diagnostics-12-00306-f017:**
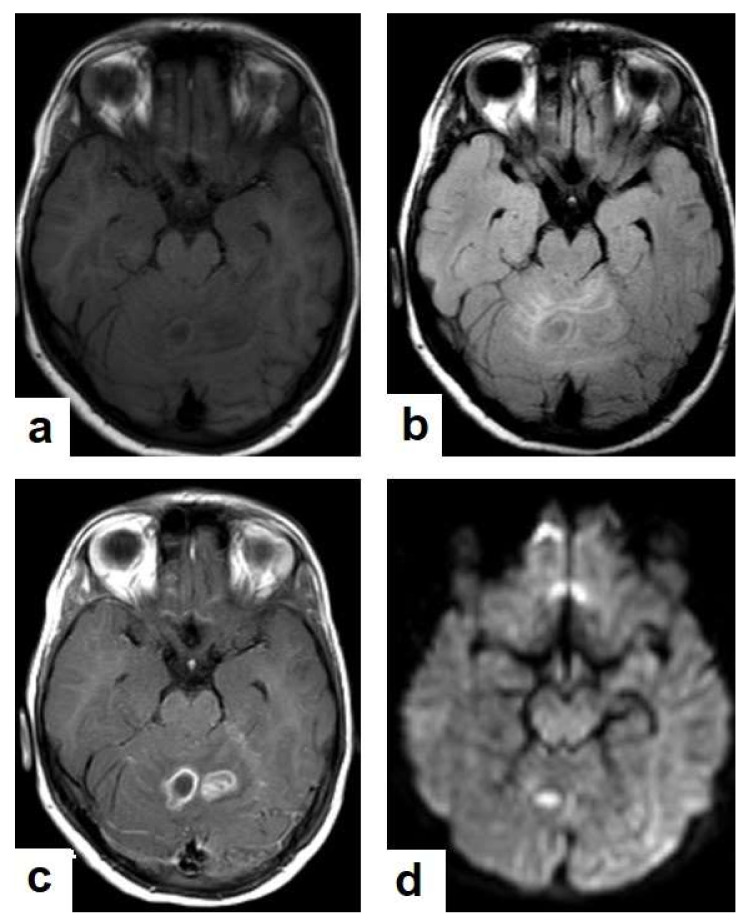
A tuberculous abscess in the cerebellum of a 13-year-old female (same patient as Figure 16). Selected images of brain MRI (**a**) T1-weighted image (WI), (**b**) FLAIR-WIs, (**c**) T1-WI with contrast, and (**d**) diffusion-weighted (DW) image. The images show a well-defined lesion in the cerebellum with low signal intensity content and a high signal intensity capsule on T1WI and FLAIR (**b**), marginal enhancement on T1WI with contrast administration (**c**), and restricted diffusion on DWI (**d**).

**Figure 18 diagnostics-12-00306-f018:**
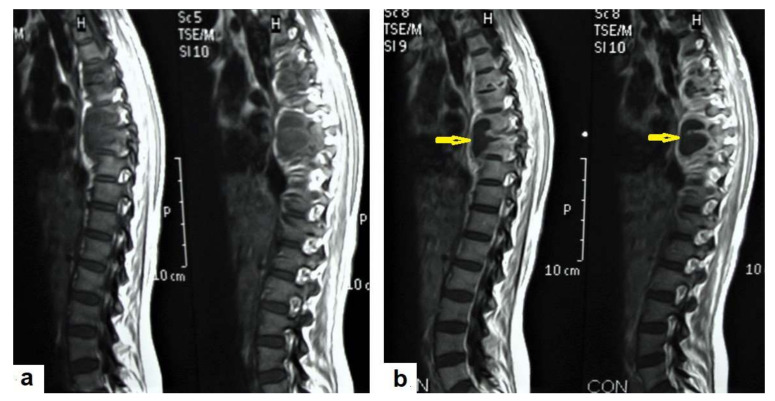
Pott’s disease in a 28-year-old female. Selected images of sagittal MRI of thoracic spine show destructive lesion affects multiple contiguous thoracic vertebrae with paraspinal collection. (**a**) Pre-contrast and (**b**) post-contrast T1-weighted images show subligamentous spread of the infection to involve five vertebrae with peripheral enhancement of the lesions, suggesting an abscess (arrow).

**Figure 19 diagnostics-12-00306-f019:**
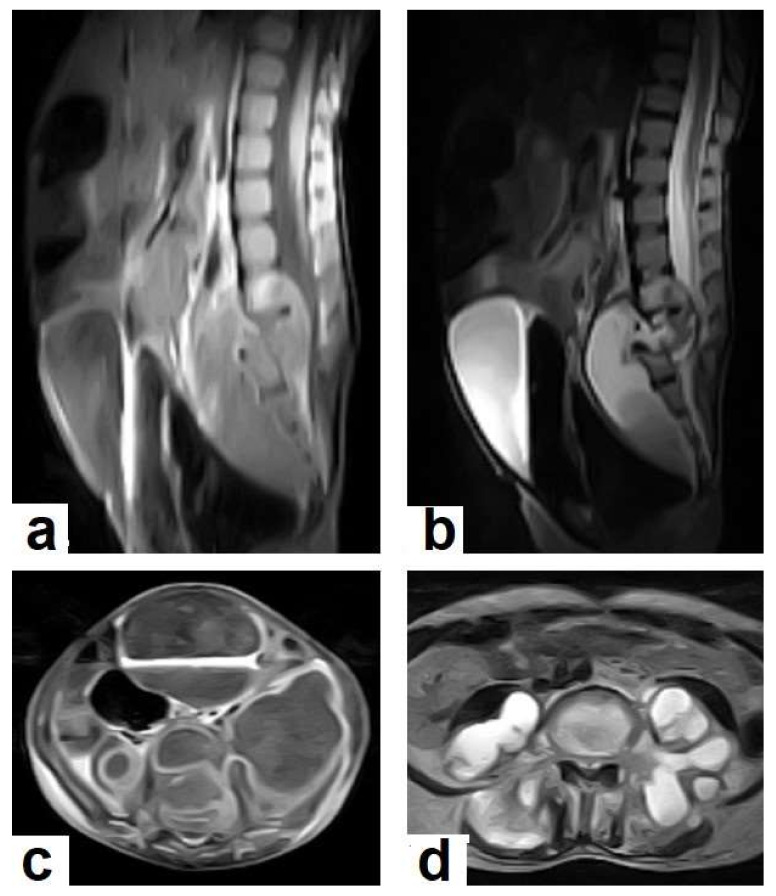
Pott’s disease in a 6-year-old female patient. Selected images of lumbosacral MRI. Sagittal (**a**) T1-weighted images and (**b**) T2-weighted images show that destructive lesion involves the T4/T5 disc and vertebrae, with extensive pre-vertebral loculated fluid collection. Selected axial images of MRI (**c**) T1-weighted image with gadolinium, and (**d**) T2-weighted image shows bilateral paravertebral loculated fluid collections involved in bilateral psoas; paravertebral muscles appear to have low-signal-intensity contents on T1-WIs with marginal enhancement after contrast administration and high-signal-intensity contents on T2-WIs (arrows). The picture is typical of Pott’s disease with bilateral psoas and paravertebral abscesses.

**Figure 20 diagnostics-12-00306-f020:**
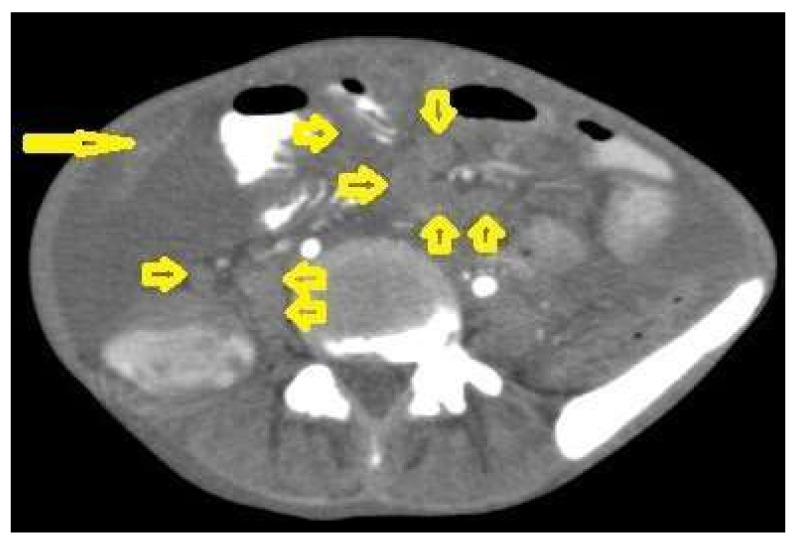
Tuberculous lymphadenitis in a 27-year-old women with cachexia, loss of appetite and cough. Axial abdominal CT image shows enlarged mesenteric, periaortic, and portahepatis lymph nodes (LNs), (short arrows) due to TB, which usually involves multiple groups, such as mesenteric and upper paraortic LNs. The image also shows relatively dense ascites and remarkable omental thickening forming cake-like mass (long arrow).

**Figure 21 diagnostics-12-00306-f021:**
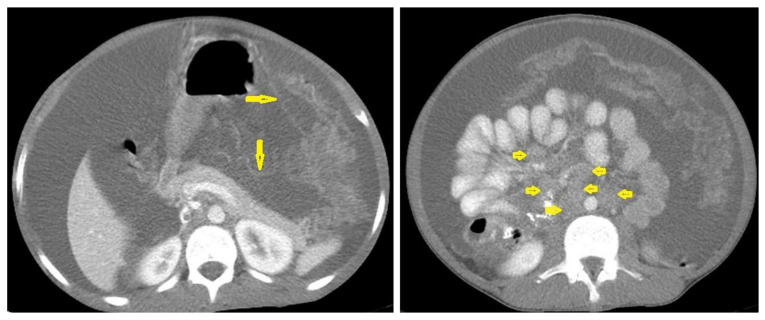
Wet peritonitis in a 16-year-old man with cachexia and loss of appetite. Selected axial abdominal images show marked relatively dense ascites and gross thickened omentum with faint enhancement of peritoneal reflections (long arrows) with multiple enlarged mesenteric and upper paraortic conglomerated lymph nodes (short arrows) with slight homogeneous enhancement.

**Figure 22 diagnostics-12-00306-f022:**
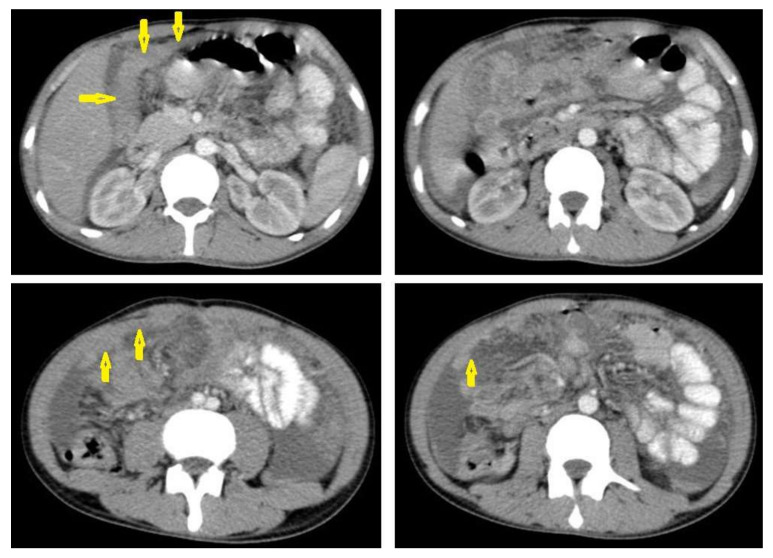
Fibrotic peritonitis in a 20-year-old man with fatigue, abdominal distension, and loss of appetite. Selected axial images of abdominal CT show remarkable omental thickening, forming cake-like masses (arrows) with faint enhancement and mild smooth thickening of peritoneal reflections associated with mild-to-moderate dense ascites and several low-attenuation mesenteric lymph nodes that are challenging to be distinguished from small bowel loops.

**Figure 23 diagnostics-12-00306-f023:**
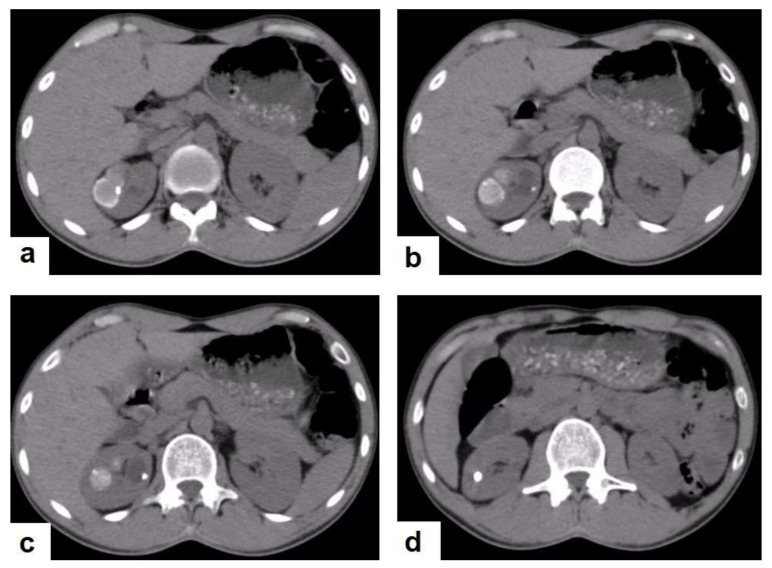
Renal TB in a 39-year-old man. Selected axial computed tomography images show (**a**) focal hyperdense non-enhancing nodules, with the largest at 26 mm, with calcified rim in the upper pole of the right kidney with focal scarring of the kidney. (**b**,**c**) Approximately, 20 mm cystic lesion in the upper calyx with focal calcification, most likely dilated calyx; and (**d**) 11 mm focal parenchymal calcification in the middle calyx.

**Figure 24 diagnostics-12-00306-f024:**
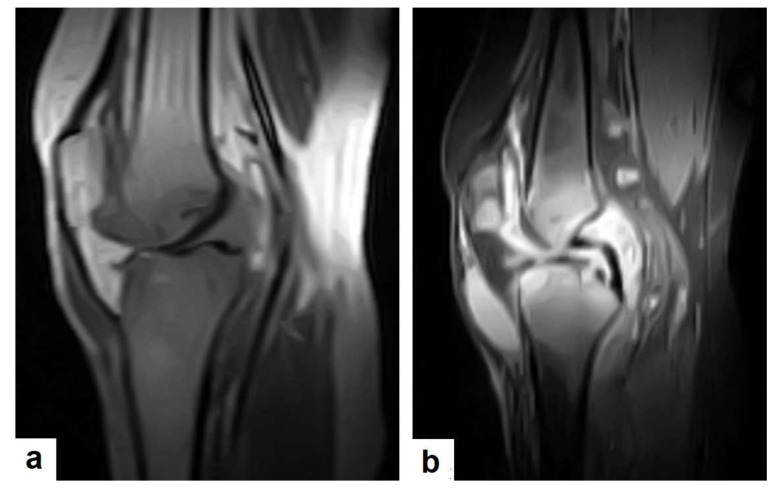
Knee TB in a 45-year-old patient with proved case of knee TB infection. Selected sagittal MRI images of (**a**) T1-weighted image and (**b**) T2-weighted image show extensive oedema of the articular surfaces of the knee, appearing as low signal intensity on T1 and high signal intensity on T2 with diffuse synovial thickening and large bone erosions (arrow heads), with mild joint effusion and a prepatellar pocket of fluid collection, which appear as low signal intensity on T1 and high signal intensity on T2. In addition, multiple enlarged popliteal lymph nodes were present.

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
