# Peer review of "The Diagnostic Deceiver: Radiological Pictorial Review of Tuberculosis"

_diagnostics, 2022, doi:10.3390/diagnostics12020306_

Round 1
Reviewer 1 Report
GENERAL COMMENTS
This is a review paper on imaging of Tb involving various systems and organs. The text is updated. The main limitation of the manuscript is the low quality of images. Given the aim of the manuscript which is to provide a pictorial essay, this is a major drawback. The use of English needs improvement throughout the text.
SPECIFIC COMMENTS
The information that mediastinal lymphadenopathy in children can cause atelectasis is missing. The arrows in fig. 1 are very thick. The image is taken with photo (unfocused), not from DICOM data.
Fig. 4 is not in black and white. b,c,d are unfocused. Please replace.
Same comment for fig. 6. Please replace.
Fig. 12,13,14,19,25. Very low quality.
Fig. 20 can be deleted as is the same as in fig. 8b.
TB spondylitis is mentioned twice, p. 17 and p. 24.
Author Response
Dear reviewer
I would like to thank you for your great effort to review and improve our manuscript. All of the possible corrections were done.
Thank you
Dear reviewer,
We would like to thank you for yours great effort to revise and your valuable advices to improve our manuscript. We have done all of the possible corrections as the following:
|
|
Reviewer comment |
Authors reply |
Section |
|
1 |
The information that mediastinal lymphadenopathy in children can cause atelectasis is missing. The arrows in fig. 1 are very thick. The image is taken with photo (unfocused), not from DICOM data. |
This information was added. The arrows were changed and colors were removed. |
Figure-1 |
|
2 |
Fig. 4 is not in black and white. b,c,d are unfocused. Please replace |
Figure was replaced by the same images without colour. |
Figure-4 |
|
3 |
Same comment for fig. 6. Please replace |
Figure was replaced by the same images without colour |
Figure-6 |
|
4 |
Fig. 12,13,14,19,25. Very low quality |
We tried to improve clarity of the images |
Figures 12, 13, 14, 19 |
|
5 |
Fig. 20 can be deleted as is the same as in fig. 8b |
Figure-20 was deleted and the arrangement of figures 21, 22, 23, 24. and 25 in the text and figures was change accordingly. |
|
|
6 |
TB spondylitis is mentioned twice, p. 17 and p. 24. |
TB spondylitis is involved in both CNS and MSK systems. |
|
Corresponding author:
Sultan Abdulwadoud Alshoabi
Associate Prof of Radiology
Taibah University
Kingdom of Saudi Arabia
10 Jan 2021
Reviewer 2 Report
In this study, Sultan AA et al., described the more common radiologic patterns of pulmonary and extrapulmonary TB. Although it is well knowledge, as the authors propose, this article could be useful for radiologists, medical students and chest physicians who are interested in the diagnosis of TB.
From my view' point this manuscript provides important information in the tuberculosis field, and it is necessary mainly in this COVID time because TB cases increased in the last WHO report, however, the current manuscript needs some changes.
Mayor comments
- Why is the references section has not numbered? I cannot see what is the numbers 1, 2, 3 etc.
- Although references have not been numbered, also I did not see the more recent TB report by WHO (2021). Please add it.
- In the introduction section where you discussed diagnostic techniques you can include the recent review by Flores J et al (Front Microbiol. 2021 Apr 15;12:638047. doi: 10.3389/fmicb.2021.638047.)
- Add bioethical information (full name of the committee that authorised, number of protocol and if you have the writing consent of patients to the use of their images).
- In the conclusion section, you can include a small phrase to “take-home” about your radiological description, because the current form of conclusion is not highlighted what you want that the future readers take as more important from your article.
Minor comments
- “mycobacterium” should be replaced by “Mycobacterium”, because the scientific names should be written with italics and the first letter in capital.
- Line 48 authors comment “…repeated exposure to mycobacterium”. I think should be “Mycobacterium tuberculosis” or authors means by exposure to the entire mycobacterium complex. Please clarify.
Author Response
Dear reviewer
I would like to thank you for your great effort in revising and suggestions to improve our manuscript. All of the possible corrections were done as in the uploaded file.
Thank you
Dear reviewer,
We would like to thank you for yours great effort to revise and your valuable advices to improve our manuscript. We have done all of the possible corrections as the following:
|
|
Reviewer comment |
Authors reply |
Section |
|
1 |
Why is the references section has not numbered? I cannot see what is the numbers 1, 2, 3 etc. |
All references are numbered with the correct order both in the text and in the references section. |
|
|
2 |
Although references have not been numbered, also I did not see the more recent TB report by WHO (2021). Please add it. |
In our scientific writing, we usually avoid to cite websites because of the information on websites can be updated or deleted at any time. TB report of WHO is available on the website of WHO. |
|
|
3 |
In the introduction section where you discussed diagnostic techniques you can include the recent review by Flores J et al (Front Microbiol. 2021 Apr 15;12:638047. doi: 10.3389/fmicb.2021.638047.) |
This was added as Reference 7 with some additions to the introduction. |
References Ref. No 7 |
|
4 |
Add bioethical information (full name of the committee that authorised, number of protocol and if you have the writing consent of patients to the use of their images). |
This is a review article and ethical approval is not applicable. Due to the retrospective collection of images. Patient consents for using images were waived. |
|
|
5 |
In the conclusion section, you can include a small phrase to “take-home” about your radiological description, because the current form of conclusion is not highlighted what you want that the future readers take as more important from your article. |
We add the following phrase at the end of conclusion “This review highlights the frequent radiological manifestations of TB which should be known by any physician.”
|
Conclusion |
|
Minor Comments: |
|||
|
1 |
“mycobacterium” should be replaced by “Mycobacterium”, because the scientific names should be written with italics and the first letter in capital. |
This was done |
All over the article |
|
2 |
Line 48 authors comment “…repeated exposure to mycobacterium”. I think should be “Mycobacterium tuberculosis” or authors means by exposure to the entire mycobacterium complex. Please clarify. |
This was corrected. |
Introduction |
Corresponding author:
Sultan Abdulwadoud Alshoabi
Associate Prof of Radiology
Taibah University
Kingdom of Saudi Arabia
10 Jan 2021